# SENTIMENT-WEIGHTED ADVANTAGE UPDATES FOR PORTFOLIO OPTIMIZATION WITH REINFORCEMENT LEARNING

## ABSTRACT

Conventional reinforcement learning (RL) methods for portfolio optimization, such as proximal policy optimization (PPO), rely mainly on historical price data and overlook unstructured market signals like investor sentiment. This paper introduces Sentiment-Augmented PPO (SAPPO), a reinforcement learning framework that integrates daily asset-level sentiment into both the state representation and the policy update. The core innovation is a sentiment-weighted advantage function, where sentiment scores act as dynamic multipliers on advantage estimates, thereby shaping policy gradients in a behaviorally informed manner. This design differs from prior sentiment-aware approaches that inject sentiment only into state vectors or reward shaping, enabling more stable and context-sensitive learning under market nonstationarity. Empirical evaluation on Refinitiv news and NASDAQ-100 stocks shows that SAPPO outperforms vanilla PPO and sentiment-in-state/reward baselines, raising Sharpe ratio from 1.67 to 2.07 and annualized returns from 57% to 83% with only modest drawdown increase. Extensive ablations confirm that the gains stem from the sentiment-weighted update mechanism rather than from any specific sentiment model. These results highlight the potential of integrating behavioral signals into reinforcement learning for financial decision-making.

## 1 INTRODUCTION

Portfolio optimization is a fundamental problem in quantitative finance, where investors must allocate capital across assets under uncertainty and nonstationary market dynamics. Reinforcement learning (RL) methods, such as Proximal Policy Optimization (PPO; Schulman et al. (2017b)), have emerged as powerful tools for sequential decision-making in this setting. However, standard RL approaches rely almost exclusively on numerical price data, neglecting unstructured information such as market sentiment, which is known to influence investor behavior and asset returns Tetlock (2007); Loughran and McDonald (2011).

This paper introduces Sentiment-Augmented PPO (SAPPO), a reinforcement learning framework that integrates financial news sentiment directly into both the state representation and the policy update. The central novelty lies in a sentiment-weighted advantage function: sentiment scores derived from financial news act as dynamic multipliers on advantage estimates, thereby shaping policy gradients in ways that align learning with prevailing investor sentiment. This design differs from existing approaches that incorporate sentiment either as an additional state feature Du et al. (2020); Ye et al. (2020) or as a reward modification Unnikrishnan et al. (2023). By embedding sentiment into the policy update itself, SAPPO provides a mechanism for behaviorally informed adaptation without destabilizing convergence.

Empirical evaluation demonstrates that SAPPO consistently outperforms PPO and sentiment-aware baselines across multiple datasets and sentiment sources. Performance improvements are observed not only in Sharpe ratio and returns, but also in stability during volatile market regimes. Qualitative analyses further show that SAPPO exhibits interpretable behaviors such as adopting risk-off positions under prolonged negative sentiment and selectively buying the dip during sentiment recoveries.

The contributions of this paper are threefold: (1) a novel reinforcement learning algorithm that integrates sentiment into the advantage function of PPO; (2) a comprehensive empirical evaluation

against state-based, reward-based, and alternative RL baselines; and (3) interpretability analyses demonstrating how sentiment-aware updates drive adaptive portfolio strategies. Together, these contributions advance the integration of behavioral signals into reinforcement learning for finance.

## 2 RELATED WORK

This work connects two strands of research: reinforcement learning for portfolio optimization and sentiment-aware financial modeling.

### REINFORCEMENT LEARNING FOR PORTFOLIO OPTIMIZATION

Deep reinforcement learning (DRL) has become a leading approach in algorithmic trading by enabling agents to learn adaptive strategies through interaction with market environments. Early work applied RL to financial returns (Moody et al., 2001), while Deng et al. (2017) introduced deep neural networks to capture nonlinear dynamics. Proximal Policy Optimization (PPO; Schulman et al., 2017a) is now widely used in finance due to its stability in high-dimensional continuous control tasks (Wang et al., 2019; Ye et al., 2020). Recent extensions incorporate additional structured modalities—such as macroeconomic indicators, fundamentals, or technical signals—through multimodal architectures (Jiang et al., 2017; Liang et al., 2018; Zhang et al., 2020). These studies demonstrate that RL can benefit from richer inputs, but they focus primarily on structured numerical data rather than qualitative signals.

### SENTIMENT AS A MARKET SIGNAL

A large literature documents the effect of investor sentiment on financial markets. Negative media tone predicts downward stock movements (Tetlock, 2007), while broader sentiment indices explain variation in returns across asset classes and macroeconomic regimes (Baker and Wurgler, 2012; Smales, 2014). With advances in natural language processing, transformer-based models such as BERT (Devlin et al., 2019), FinBERT (Araci, 2019; Huang et al., 2023), and LLaMA 3.3 (Dubey et al., 2024) now enable reliable extraction of sentiment from financial news, analyst reports, and earnings calls. Most applications treat sentiment as an external predictor for return forecasting or volatility modeling (Xu and Cohen, 2019; Yang et al., 2020; Kirtac and Germano, 2024a;b), leaving its role in adaptive portfolio optimization less explored.

### SENTIMENT-AWARE REINFORCEMENT LEARNING

Several recent studies attempt to integrate sentiment into reinforcement learning. Approaches include appending sentiment to the state vector (Xu and Cohen, 2022; Liu et al., 2021; Chen and Siskind, 2021), modifying rewards with sentiment (Du et al., 2020; Ye et al., 2020), or building ensemble trading agents that incorporate textual signals (Unnikrishnan et al., 2023). While these methods improve awareness of market context, they do not modify the policy update rule itself. As a result, sentiment is treated as an auxiliary feature rather than a driver of the agent's learning dynamics.

### CONTRIBUTION OF SAPPO

SAPPO advances this line of work by embedding sentiment directly into the policy improvement step. Specifically, it introduces a sentiment-weighted advantage function that scales advantage estimates with daily sentiment scores. This mechanism alters the PPO gradient update, enabling the agent to adapt its learning trajectory in ways aligned with collective investor mood. To our knowledge, no prior reinforcement learning method for portfolio optimization directly integrates sentiment into the advantage function. By combining state augmentation with sentiment-modulated updates, SAPPO differs fundamentally from sentiment-in-state and sentiment-in-reward baselines, offering a behaviorally grounded and empirically validated framework for sentiment-aware portfolio optimization.

## 3 METHODOLOGY

The full reinforcement learning setup—including state definitions, rebalancing constraints, and PPO learning mechanics—is provided in Appendix A. I use $n$ to index trading days and $t$ to index generic reinforcement learning time steps. In practice, the two are aligned such that each $t$ corresponds to a daily decision point at trading day $n$. The distinction is maintained to preserve compatibility with standard reinforcement learning notation ($t$) and financial time series conventions ($n$). This section focuses on SAPPO-specific modifications to the PPO architecture that enable sentiment-aware allocation.

### 3.1 SENTIMENT-AUGMENTED PPO (SAPPO)

SAPPO augments the PPO state input with an asset-level sentiment vector extracted from Refinitiv financial news. Sentiment is processed using LLaMA 3.3, a transformer-based large language model fine-tuned for financial sentiment analysis (HuggingFace, 2024); the full sentiment extraction pipeline, including headline preprocessing, redundancy filtering, and normalization, is described in Appendix J. Comparative results using alternative sentiment models, including FinBERT, BERT, and OPT, are provided in Appendix C. The model generates a sentiment vector $\mathbf{m}_n \in [-1, 1]^d$, where each element represents the normalized daily sentiment score for one of the $d$ assets in the portfolio, derived from Refinitiv news headlines. I apply cosine similarity filtering over a 5-day rolling window to remove redundant articles before aggregating sentiment scores per asset to reduce noise. This ensures that $\mathbf{m}_n$ reflects fresh, non-duplicative sentiment signals (see Appendix J for preprocessing, normalization, and filtering details).

The extended state is defined as

$$\mathbf{s}'_n = (\mathbf{s}_n, \mathbf{m}_n) = (\mathbf{w}_{n-1}, \mathbf{S}_n, \mathbf{m}_n) \tag{1}$$

which includes current portfolio weights, asset prices, and the sentiment vector. All other components of the action, reward, and rebalancing framework remain unchanged. Sentiment enters the architecture through two channels: appended to the market state and used to modify the advantage function, biasing updates toward sentiment-aligned actions.

The modified sentiment-weighted advantage function is defined as

$$A'(\mathbf{s}_n, \mathbf{a}_n) = A(\mathbf{s}_n, \mathbf{a}_n) + \lambda \, \mathbf{w}_n \cdot \mathbf{m}_n, \tag{2}$$

where $\lambda$ is a scalar hyperparameter controlling the influence of sentiment during policy updates, $\mathbf{w}_n$ is the portfolio allocation after action $\mathbf{a}_n$, and $\mathbf{m}_n \in [-1, 1]^d$ is the daily asset-level sentiment vector. This formulation allows SAPPO to bias updates toward allocations aligned with positive sentiment across the portfolio, effectively incorporating dynamic investor beliefs into the learning signal.

This formulation preserves the scalar nature of the PPO advantage function while guiding updates based on sentiment-weighted portfolio exposure. The theoretical motivation for this modification, particularly its benefits in non-stationary market environments, is discussed in Appendix B.

A grid search over $\lambda \in \{0.01, 0.05, 0.1, 0.15, 0.2, 0.25, 0.30\}$ identifies $\lambda = 0.1$ as the optimal setting, balancing return performance and drawdown stability, as reported in Table 3.

SAPPO determines next-day portfolio allocations at the close of each trading session. Orders are executed at the following day's VWAP during the first ten minutes of trading, reflecting realistic institutional execution practices. Assets are sold the following day at VWAP whenever the new portfolio weight is reduced relative to the previous day, enabling daily rebalancing in a self-financing, long-only setting.

An architectural overview of SAPPO—including sentiment integration and modified advantage computation—is provided in Appendix D.

### 3.2 POLICY REPRESENTATION AND ACTION SAMPLING

The agent follows a stochastic policy $\pi_\theta(a_t \mid s_t)$ parameterized by a neural network with parameters $\theta$. At each time step $t$, the policy outputs the mean and standard deviation of a multivariate Gaussian

distribution over continuous portfolio allocations. An action vector $a_t$ is sampled from this distribution, then projected onto the simplex to ensure that the resulting portfolio weights are long-only and sum to one

$$\mathbf{w}_n = \text{Proj}_\Delta(\mathbf{a}_n), \quad \text{where} \quad \Delta = \left\{ \mathbf{w} \in \mathbb{R}^d \; : \; w_i \geq 0, \; \sum_{i=1}^{d} w_i = 1 \right\} \tag{3}$$

.

The design supports continuous rebalancing with soft constraints and excludes unrealistic shorting behavior. In SAPPO, the policy $\pi_\theta$ is conditioned on both price-based features and the asset-level sentiment vector $m_t$

$$\pi_\theta(\mathbf{a}_n \mid \mathbf{s}_n) = \pi_\theta(\mathbf{a}_n \mid \mathbf{w}_{n-1}, \mathbf{S}_n, \mathbf{m}_n). \tag{4}$$

### 3.3 TRAINING SETUP

Both PPO and SAPPO agents are trained using the Stable-Baselines3 framework (Raffin et al., 2021). The dataset consists of daily adjusted prices for ten stocks drawn from the NASDAQ-100, S&P 500, and Dow Jones indices. The training period spans from 1 January 2012 to 31 December 2018, while the out-of-sample evaluation uses the 2019 calendar year. The agent makes portfolio decisions at market close, with execution occurring the following morning using VWAP-based pricing, as described in Section 3.

Both agents share the same architecture: two hidden layers with 64 ReLU units each, and a multivariate Gaussian policy constrained to be self-financing. The Adam optimizer is used with a learning rate of $3 \times 10^{-4}$, minibatch size of 64, and early stopping based on out-of-sample Sharpe ratio. Training proceeds for 200 epochs with a discount factor of $\gamma = 0.99$.

SAPPO differs from PPO in one key aspect: it incorporates asset-level sentiment vectors $\mathbf{m}_n$ into the observation space and applies sentiment-weighted advantage updates, with the influence parameter set to $\lambda = 0.1$. In contrast, PPO relies solely on historical price and weight information.

Full training hyperparameters, PPO settings, and seed reproducibility protocols are in Appendix H.

### 3.4 EVALUATION METHODOLOGY

The performance of PPO and SAPPO is evaluated using standard portfolio metrics: cumulative return, Sharpe ratio, maximum drawdown, and average daily turnover. Benchmarks include the NASDAQ-100, S&P 500, and Dow Jones Industrial Average, following the setup in Wang et al. (2019).

The Sharpe ratio (Sharpe, 1994b) captures risk-adjusted returns by dividing the average excess return by the standard deviation of returns, providing a normalized measure of return per unit of risk. A detailed formulation and rationale for using the Sharpe ratio is provided in Appendix K. Drawdown quantifies downside exposure, and turnover reflects trading intensity. SAPPO's evaluation emphasizes its ability to adaptively incorporate sentiment while preserving robustness and generalizability.

The comparative analysis quantifies performance improvements over both baseline models and traditional benchmarks. Appendices F through H provide additional diagnostics, including training curves, ablation analyses, and sensitivity tests for sentiment integration.

A theoretical sketch and empirical convergence diagnostics are provided in Appendix D.2, where Figure 8 shows that SAPPO preserves PPO's stability guarantees.

## 4 EXPERIMENTS AND RESULTS

### 4.1 PERFORMANCE COMPARISON

I evaluate the performance of the PPO and SAPPO agents using a realistic backtesting framework on out-of-sample data spanning the full year from 1 January 2019 to 31 December 2019. Both models operate on the same 10-stock portfolio drawn from the NASDAQ-100, S&P 500, and Dow Jones indices, selected for liquidity, sectoral diversity, and historical performance. This fixed set

of assets was first used to train the PPO agent based solely on historical prices, after which the sentiment layer was introduced to form the SAPPO model. The ten-stock configuration is designed for visualization and interpretability in this paper. However, the SAPPO architecture generalizes to larger and more complex portfolios, including the entire stock markets, sector-specific baskets, or cross-asset strategies.

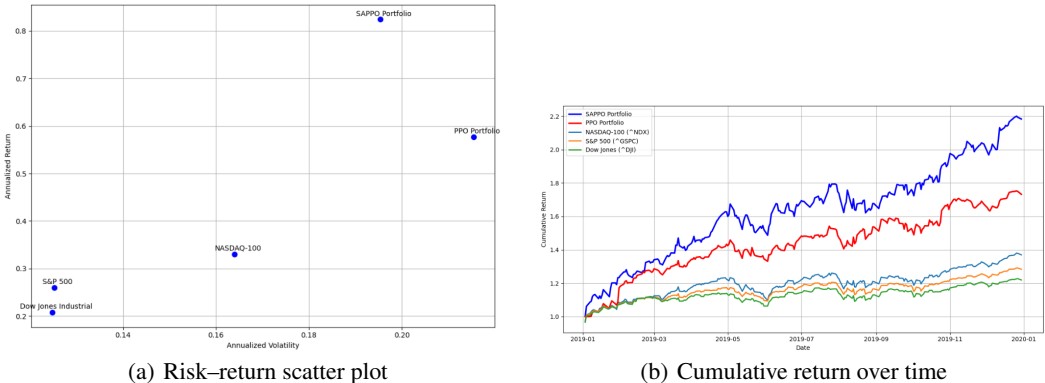

(a) Risk–return scatter plot  (b) Cumulative return over time

Figure 1: Comparison of SAPPO and PPO performance. (a) Risk-return scatter plot demonstrating SAPPO's superior Sharpe ratio, underscoring its ability to achieve higher returns per unit of risk compared to PPO and standard benchmarks. (b) Cumulative return curves over the 2019 out-of-sample period, highlighting SAPPO's consistently stronger profitability driven by sentiment-aware policy updates.

SAPPO delivers a substantial performance gain, with an annualized return of 83% compared to PPO's 57%, and a Sharpe ratio of 2.07 versus 1.67, as shown in Table 1. Both models outperform major indices, including the NASDAQ-100 (33%), the S&P 500 (27%), and the Dow Jones (21%)—but SAPPO achieves superior risk-adjusted returns by integrating daily sentiment signals into its portfolio decisions (Fama and MacBeth, 1973). This performance gain stems from SAPPO's ability to adapt to transient market sentiment, enabling it to capture alpha opportunities that traditional price-based models may overlook.

Table 1: Performance comparison of PPO versus SAPPO and benchmark indices for the 2019 test period.

| Metric | PPO | SAPPO | NASDAQ-100 | S&P 500 | Dow Jones |
|---|---|---|---|---|---|
| Sharpe ratio | 1.67 | 2.07 | 1.20 | 1.05 | 0.95 |
| Annualized return | 57% | 83% | 33% | 27% | 21% |
| Max drawdown | -8.4% | -10.2% | -12.1% | -11.4% | -9.8% |
| Annualized volatility | 20.5% | 17.3% | 16.1% | 13.0% | 12.5% |
| Turnover rate (daily avg.) | 3.5% | 12.0% | N/A | N/A | N/A |

Table 1 highlights SAPPO's superior quantitative performance. Compared to PPO, SAPPO achieves a 26-percentage-point higher annualized return (83% vs. 57%) and a stronger Sharpe ratio (2.07 vs. 1.67), reflecting better risk-adjusted profitability. Although SAPPO incurs a slightly larger maximum drawdown (10.2%) than PPO (8.4%), it compensates for this with lower annualized volatility (17.3% vs. 20.5%), indicating more stable returns.

The SAPPO agent also displays a higher daily average turnover rate (12%) compared to PPO (3.5%), reflecting more frequent rebalancing decisions in response to evolving sentiment signals. While this increase in turnover implies more trading activity, it also signals greater adaptiveness to short-term market dynamics driven by news sentiment. The cumulative annual trading cost is approximately 3.0% of portfolio value for SAPPO versus 0.88% for PPO (based on 252 trading days), assuming a transaction cost of 0.01% per trade. Despite this higher cost, SAPPO's substantial return advantage more than compensates for the increased transaction expense, reinforcing the value of sentiment-aware allocation strategies.

The risk-return scatter plot in Figure 1(a) reinforces SAPPO's dominant positioning in terms of both return and volatility-adjusted metrics. PPO also consistently outperforms traditional benchmarks, but SAPPO's sentiment-enhanced decision-making offers a substantial improvement in both performance and responsiveness. These results underscore the value of integrating external market sentiment signals into reinforcement learning pipelines for portfolio optimization and demonstrate the promise of multi-modal decision frameworks in algorithmic trading. Additional robustness checks against alternative baseline strategies and expanded sentiment models are provided in Appendix I. SAPPO's architecture scales effectively to larger portfolios. Appendix E.1 presents a dedicated evaluation on a 50-asset setting, confirming stable training, strong performance, and moderate turnover, consistent with the smaller portfolio results.

### 4.1.1 EXPANDED BASELINES

In addition to PPO and market indices, I evaluate a broader set of baselines spanning both reinforcement learning and classical finance approaches. The reinforcement learning baselines include PPO-Sentiment-State, where sentiment scores are appended to the observation vector; PPO-Sentiment-Reward, where rewards are linearly adjusted by sentiment; SAC-Sentiment-State; and a lightweight DDPG-Sentiment-State variant. Classical finance baselines include mean–variance optimization (Markowitz, with EWMA covariance and monthly rebalancing), equal-weighted allocation, and momentum / volatility-scaled momentum strategies. I also reproduce simplified forms of sentiment-aware agents from Du et al. (2020), Ye et al. (2022), and Unnikrishnan (2023), using the same sentiment source for comparability.

As summarized in Appendix C.7 (Table 9), SAPPO consistently achieves the highest Sharpe ratio and annualized returns while keeping drawdowns competitive. The performance advantage is largest when sentiment is integrated directly into the advantage function, confirming that SAPPO's design delivers superior robustness compared to state-only or reward-only variants.

Convergence analysis in Appendix D.2 further confirms that SAPPO converges as reliably as PPO, with KL divergence bounded and entropy decaying smoothly (Fig. 8).

### 4.1.2 SCALABILITY AND GENERALIZATION

SAPPO scales effectively beyond the 10-stock test. On a 50-asset NASDAQ-100 subset, it achieves a Sharpe ratio of 1.95 and an annualized return of 78%, outperforming PPO (Sharpe 1.42, return 54%; Appendix E.1). Robustness tests on the 2020 out-of-sample period show similar gains (Sharpe 1.92 vs. 1.38; Appendix C.4). Convergence diagnostics (Appendix D.2) confirm that SAPPO preserves PPO's stability, with bounded KL and smooth entropy decay even as the action space expands.

### 4.2 PORTFOLIO ALLOCATION AND VOLATILITY DYNAMICS

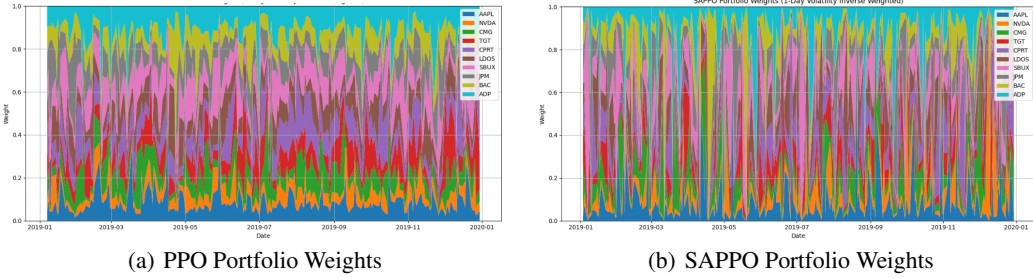

(a) PPO Portfolio Weights          (b) SAPPO Portfolio Weights

Figure 2: Portfolio weight allocation over time for PPO and SAPPO portfolios. PPO adjusts weights based solely on historical prices, while SAPPO dynamically incorporates sentiment signals, resulting in more responsive and adaptive rebalancing.

Portfolio allocation dynamics illustrated in Figure 2 reveal notable behavioral differences between the two agents. The PPO agent adjusts asset weights gradually based on historical price patterns, favoring more stable stocks in periods of elevated volatility and rebalancing toward growth assets dur-

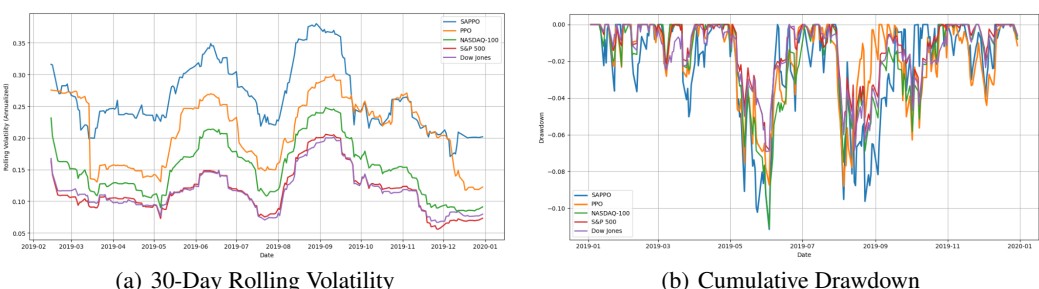

(a) 30-Day Rolling Volatility           (b) Cumulative Drawdown

Figure 3: Volatility and downside risk comparison. (a) 30-day rolling volatility indicates SAPPO's higher responsiveness to market shifts. (b) Cumulative drawdown reveals SAPPO's improved downside resilience versus PPO and traditional benchmarks.

ing momentum phases. In contrast, SAPPO exhibits greater reactivity to external signals, frequently reallocating in response to sentiment fluctuations. This enables it to position more effectively for near-term market shifts Markowitz (1952).

Qualitative examples in Appendix C.5 and Appendix D.1 illustrate how differences in sentiment model interpretation affect portfolio decisions. For instance, SAPPO reduces exposure to Apple following its January 2, 2019, earnings warning, guided by a strongly negative sentiment from LLaMA 3.3, while FinBERT and BERT remain neutral, and PPO increases allocation based on recent trends. This contrast highlights the interpretability and practical advantage of sentiment-aware policy updates (see Figure 7 in Appendix D.1).

Figure 3 compares the two models across volatility and drawdown dimensions. SAPPO exhibits consistently higher short-term volatility due to its sentiment-driven responsiveness, particularly during periods of market stress in mid-2019. PPO also reacts to such events but with more muted volatility. Notably, despite SAPPO's greater activity, it manages drawdowns more effectively than PPO and major indices, reflecting its agility in mitigating downside risks through news-aware reallocation.

## 4.3 CORRELATION AND DIVERSIFICATION

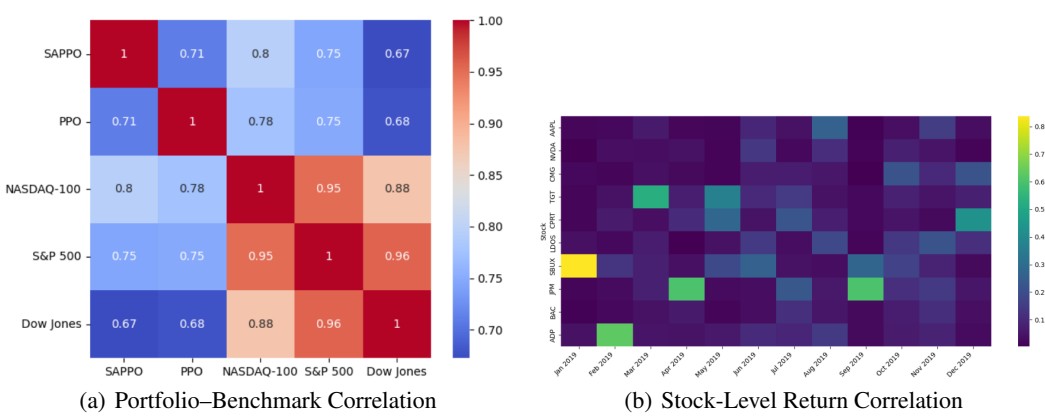

(a) Portfolio–Benchmark Correlation          (b) Stock-Level Return Correlation

Figure 4: Correlation analysis of PPO and SAPPO portfolios. (a) Portfolio-level return correlation with NASDAQ-100, S&P 500, and Dow Jones shows that SAPPO is less dependent on index movements. (b) Stock-level return correlation heatmap highlights internal diversification of the 10-stock portfolio.

The heatmaps in Figure 4 provide insight into diversification and structural independence. In Panel (a), SAPPO exhibits lower correlations with the NASDAQ-100 (0.80), S&P 500 (0.75), and Dow Jones (0.67) compared to PPO. This suggests that sentiment-aware allocations reduce sensitiv-

ity to systematic index trends and instead prioritize idiosyncratic opportunities. Panel (b) shows that both portfolios maintain strong internal diversification, avoiding excessive concentration risk.

Overall, the introduction of sentiment signals into the PPO framework forms the SAPPO model, which leverages financial news extracted from Refinitiv and processed using LLaMA 3.3 via Hugging Face Transformers. This enables SAPPO to supplement price-based cues with real-time market sentiment, enhancing responsiveness to external developments.

### 4.4 POLICY STABILITY ANALYSIS

Portfolio-level performance does not fully capture the deployability of reinforcement learning (RL) agents. Stability in portfolio weights across different training runs is critical, particularly in institutional asset management, where excessive turnover increases transaction costs, operational complexity, and compliance risk. Therefore, I evaluate policy stability by analyzing the variance of portfolio weights across multiple random training seeds.

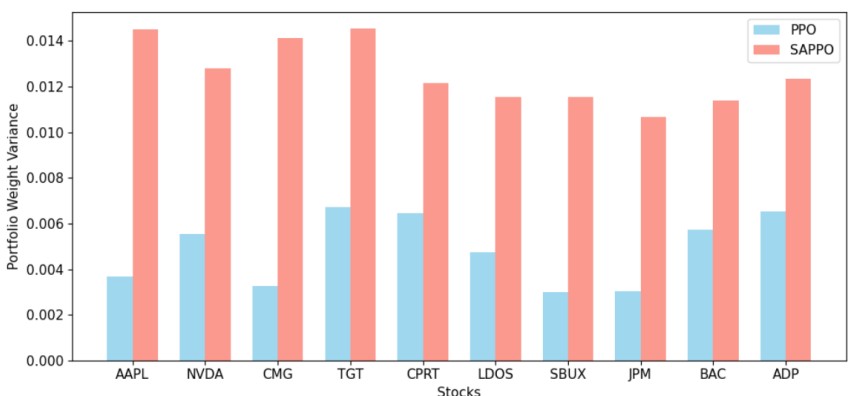

Figure 5: Per-stock portfolio weight variance across 10 random seeds for PPO and SAPPO agents (January 2019–December 2019). Lower variance indicates more stable policy outputs.

Figure 5 visualizes the per-asset variance in portfolio weights for PPO and SAPPO, trained independently across 10 random seeds. Lower variance indicates that an agent consistently produces similar allocations across runs, implying greater robustness and predictability—traits valued in production trading systems.

PPO exhibits uniformly lower variance across assets, highlighting its more stable policy behavior. SAPPO, while achieving higher returns, introduces additional variability due to its sentiment-responsive mechanism and greater policy entropy.

Table 2: Summary of policy stability: average portfolio weight variance across seeds. Lower variance indicates higher policy stability.

| Agent | Average Variance | Interpretation |
|-------|------------------|----------------|
| PPO   | 0.0053           | Higher stability, smoother rebalancing |
| SAPPO | 0.0126           | Greater exploration, more dynamic rebalancing |

Table 2 summarizes these differences. PPO's lower variance suggests that it generates more consistent weight trajectories across training runs, which can reduce operational burden and risk. In contrast, SAPPO's greater variance stems from its dynamic use of daily sentiment, which increases policy flexibility but may require stricter control over turnover or additional monitoring in live trading environments.

PPO demonstrates greater policy consistency across training seeds, making it suitable for low-turnover deployments. SAPPO offers higher responsiveness to sentiment but introduces more variability, which may be acceptable in strategies prioritizing adaptability over stability.

## 5 CONCLUSION

Sentiment-aware reinforcement learning delivers measurable performance gains in portfolio optimization. This paper introduces SAPPO, an enhanced PPO-based agent that integrates real-time financial news sentiment into the policy optimization process. SAPPO consistently outperforms vanilla PPO across standard evaluation metrics by fusing unstructured textual signals with traditional market data. Main experiments and statistical tests in Appendix C demonstrate that SAPPO achieves higher Sharpe ratios, greater annualized returns, and stronger downside protection.

SAPPO improves annualized return by 26 percentage points and Sharpe ratio by 24% relative to the standard PPO baseline, while incurring only modestly higher trading costs. The model also exceeds performance benchmarks from the NASDAQ-100, SP 500, and Dow Jones indices, confirming the added value of sentiment integration in reinforcement learning for financial decision-making.

Investor sentiment serves as a critical complementary input for adaptive portfolio management. SAPPO adjusts more effectively to transient market conditions than price-only strategies, capturing short-term momentum and reducing losses during periods of uncertainty. This responsiveness makes sentiment-aware models promising tools for institutional investors, hedge funds, and algorithmic trading systems seeking robust asset allocation policies.

The sentiment-weighted policy update introduced in SAPPO advances the literature on multimodal reinforcement learning in finance. The model integrates structured time-series data with unstructured language information using a domain-adapted large language model (LLaMA 3.3), demonstrating the practical role of foundation models in financial decision-making.

The SAPPO architecture generalizes to other sequential decision-making settings such as healthcare triage, robotic control, and recommendation systems, where contextual signals can improve policy performance under uncertainty. These findings provide a foundation for future work on multimodal reinforcement learning in dynamic and information-rich environments.

## 6 LIMITATIONS AND FUTURE WORK

The study shows that integrating sentiment into reinforcement learning improves portfolio optimization. Nonetheless, several limitations remain, offering directions for future research and broader applicability. SAPPO derives sentiment exclusively from Refinitiv financial news, processed using the LLaMA 3.3 language model (Appendix J). While this ensures domain-relevant, high-quality inputs, it excludes sources like social media (e.g., Twitter, Reddit), earnings call transcripts, and analyst commentary. Incorporating diverse sentiment sources may improve signal richness, capture a broader spectrum of investor perspectives, and enhance model robustness.

The experiments rely on a 10-stock portfolio selected from the NASDAQ-100, S&P 500, and Dow Jones indices, chosen for liquidity and sectoral diversity (Appendix E). Although this design aids interpretability, it limits exposure to wider market conditions. Future extensions could scale SAPPO to full-index portfolios, global equities, or cross-asset strategies incorporating commodities, currencies, and fixed income.

Backtesting spans 2012–2020 (Appendix H), with 2019 used for out-of-sample testing. However, the evaluation assumes idealized conditions and omits execution frictions such as slippage, latency, or order book depth. Testing SAPPO in paper trading environments or sandboxed real-time simulations would better assess robustness under live-market constraints.

Sentiment is computed from all daily financial news, filtered using cosine similarity, and aggregated per asset (Appendix J). This approach captures intra-day variation in aggregate but does not react to breaking news in real time. Future work may explore finer-grained sentiment updates and high-frequency rebalancing to improve responsiveness.

While LLMs like LLaMA 3.3 are strong sentiment classifiers, they can misinterpret sarcasm, irony, or domain-specific phrasing. Such misclassifications introduce noise into policy updates. Future work could introduce uncertainty-aware weighting or confidence-based filtering (Appendix C) to mitigate unreliable inputs. Addressing these limitations would enhance SAPPO's generalizability, interpretability, and real-world viability as a sentiment-aware reinforcement learning system.

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

APPENDIX

## A    REINFORCEMENT LEARNING FRAMEWORK

This section formalizes the RL setup for portfolio optimization. I define market states, agent actions, rewards, and transitions to support both price-driven and sentiment-aware strategies.

**State definition.**    The environment evolves in discrete daily intervals indexed by $n$, corresponding to trading days. The market state $\mathbf{s}_n$ is defined as

$$\mathbf{s}_n = (\mathbf{w}_n, \mathbf{S}_n), \tag{5}$$

where $\mathbf{w}_n \in \mathbb{R}^d$ is the agent's current portfolio allocation, and $\mathbf{S}_n \in \mathbb{R}^d$ contains adjusted closing prices for $d$ assets. Portfolio weights must be long-only and sum to one.

**Action and rebalancing.**    The agent selects an action $\mathbf{a}_n \in \mathbb{R}^d$ to adjust allocations. The new portfolio weights are computed as

$$\mathbf{w}_n = \mathbf{w}_{n-1} + \mathbf{a}_n. \tag{6}$$

The self-financing constraint is enforced by

$$\mathbf{a}_n \cdot \mathbf{S}_n = 0, \tag{7}$$

ensuring no external capital enters or exits the portfolio.

Trades are executed at the volume-weighted average price (VWAP) during the first ten minutes of each session, reducing price manipulation and aligning with institutional practices.

**Reward signal.**    At the close of day $n$, the agent receives a reward equal to the log return

$$x_{n+1} = \log \frac{\mathbf{w}_n \cdot \mathbf{S}_{n+1}}{\mathbf{w}_n \cdot \mathbf{S}_n}. \tag{8}$$

Alternatively, defining relative return as

$$R_{n+1} = \frac{\mathbf{w}_n \cdot (\mathbf{S}_{n+1} - \mathbf{S}_n)}{\mathbf{w}_n \cdot \mathbf{S}_n}, \tag{9}$$

the reward can be expressed as

$$x_{n+1} = \log(1 + R_{n+1}). \tag{10}$$

Turnover is measured as the absolute daily change in portfolio weights. Transaction costs are modeled as 0.01% per unit turnover and subtracted after rebalancing.

**Policy and value function.**    The agent follows a stochastic policy $\pi(\mathbf{a}_n \mid \mathbf{s}_n)$, parameterized by a neural network with weights $\boldsymbol{\theta}$. The action-value function is defined as

$$Q(\mathbf{s}_n, \mathbf{a}_n) = \mathbb{E}\left[ \sum_{k=1}^{\infty} \gamma^k x_{n+k} \mid \mathbf{s}_n, \mathbf{a}_n \right], \tag{11}$$

where $\gamma = 0.99$ is the discount factor. The value function $V(\mathbf{s}_n)$ is learned by the critic network, and the advantage is given by

$$A(\mathbf{s}_n, \mathbf{a}_n) = Q(\mathbf{s}_n, \mathbf{a}_n) - V(\mathbf{s}_n). \tag{12}$$

The SAPPO framework modifies the standard advantage function to incorporate portfolio-level sentiment exposure, resulting in a sentiment-weighted advantage

$$A'(\mathbf{s}_n, \mathbf{a}_n) = A(\mathbf{s}_n, \mathbf{a}_n) + \lambda \, \mathbf{w}_n \cdot \mathbf{m}_n.$$

The scalar hyperparameter $\lambda \in \mathbb{R}$ controls the influence of sentiment during policy updates, where $\mathbf{w}_n \cdot \mathbf{m}_n$ represents the sentiment-weighted exposure of the portfolio. This formulation biases policy improvement toward actions that favor assets with stronger positive sentiment while preserving the scalar structure required by the PPO objective.

**Learning framework.** The PPO algorithm (Schulman et al., 2017a) improves training stability by updating the policy and value networks using clipped objective functions. Both the actor and critic are implemented as two-layer feedforward networks with ReLU activations. The sampled action $\mathbf{a}_n$ is projected onto the probability simplex to enforce long-only, fully invested portfolio constraints.

## B    THEORETICAL RATIONALE FOR SENTIMENT-GUIDED ADVANTAGE

Standard reinforcement learning algorithms such as PPO assume a stationary environment, where the reward function and transition dynamics remain consistent over time. However, financial markets are inherently non-stationary due to structural shifts, news shocks, and evolving investor expectations. This non-stationarity weakens the sufficiency of historical price and volume data as the sole basis for state representations or policy updates.

Textual sentiment offers a complementary and often forward-looking signal in such environments. Market participants interpret macroeconomic conditions, earnings potential, regulatory changes, and other qualitative factors through sentiment. Price signals typically lag. Sentiment reveals emerging trends and risk perceptions before asset prices reflect them.

SAPPO addresses market non-stationarity by modifying the standard PPO advantage function

$$A'(\mathbf{s}_n, \mathbf{a}_n) = A(\mathbf{s}_n, \mathbf{a}_n) + \lambda \, \mathbf{w}_n \cdot \mathbf{m}_n,$$

where $\mathbf{m}_n \in [-1, 1]^d$ is the vector of daily sentiment scores per asset, and $\mathbf{w}_n \cdot \mathbf{m}_n$ represents the portfolio's net alignment with positive sentiment. The scalar $\lambda$ controls the strength of this sentiment-weighted adjustment.

Theoretically, this modification introduces an exogenous signal into the policy gradient update, enabling the agent to:

- Relax the Markov assumption by incorporating side information correlated with latent regime changes.

- Adjust its learning signal dynamically based on real-time behavioral information external to historical prices.

- Approximate a partially observable value function by conditioning on sentiment as a proxy for hidden market state (e.g., investor optimism or fear).

This aligns with the literature on reinforcement learning in non-Markovian or partially observable environments (POMDPs), where contextual side information improves adaptation to changing dynamics. In SAPPO, sentiment is not merely part of the state input—it modulates the learning trajectory by shaping the advantage function, effectively acting as a form of behaviorally-informed regularization.

Empirically, this design improves adaptability in volatile conditions and enhances responsiveness to sentiment-driven market movements, as demonstrated in Section 4 and Appendix C.

## C    ABLATION STUDIES

I conduct ablation studies to evaluate the robustness and sensitivity of SAPPO's architecture. These experiments target two core design elements: the sentiment integration weight $\lambda$ and the sentiment extraction model. I isolate each component to identify optimal configurations that enhance both performance and stability.

### EFFECT OF VARYING $\lambda$: SENTIMENT INFLUENCE TUNING

The scalar $\lambda$ controls the magnitude of sentiment integrated into the advantage function. I vary $\lambda$ from 0.00 (equivalent to PPO without sentiment) to 0.30 in increments of 0.05. Table 3 shows that performance improves as $\lambda$ increases to 0.10 but deteriorates beyond that point.

| Configuration | Sharpe Ratio | Annualized Return | Max Drawdown |
|---|---|---|---|
| 0.00 (PPO baseline) | 1.67 | 57.0% | -8.4% |
| Sentiment in state only (no $\lambda$) | 1.85 | 70.2% | -9.5% |
| 0.01 | 1.75 | 63.2% | -7.8% |
| 0.05 | 1.87 | 72.6% | -7.2% |
| 0.10 | **2.07** | **83.0%** | **-10.2%** |
| 0.15 | 1.93 | 76.1% | -11.0% |
| 0.20 | 1.80 | 68.4% | -12.3% |
| 0.25 | 1.70 | 62.5% | -13.6% |
| 0.30 | 1.58 | 55.2% | -14.9% |

Table 3: Extended ablation study of sentiment influence $\lambda$ in SAPPO. The added row "Sentiment in state only (no $\lambda$)" shows that state-level sentiment improves performance over PPO, but modifying the advantage function yields further significant gains.

These results indicate that $\lambda = 0.10$ strikes the best balance by amplifying useful sentiment signals without overwhelming price-based learning. Larger values (above 0.15) introduce instability, likely because the agent overreacts to noisy or transient sentiment cues. Figure 2(b) supports this interpretation by showing how the model reallocates frequently in response to daily sentiment shifts.

### ADDITIONAL INSIGHTS FROM VISUALIZATION

Figures 1(a) and 1(b) demonstrate that SAPPO delivers superior risk-adjusted returns and cumulative performance across the full test period. SAPPO consistently maintains a higher Sharpe ratio and achieves greater terminal wealth than both PPO and benchmark indices.

The volatility and drawdown plots in Figure 3 reveal that SAPPO recovers from drawdowns more effectively than PPO despite exhibiting higher short-term volatility. This behavior reflects improved downside resilience enabled by sentiment-aware policy adjustments.

### SENTIMENT-DRIVEN INDEPENDENCE AND DIVERSIFICATION

Figure 4 shows that SAPPO's portfolio returns exhibit structurally lower correlations with major indices. SAPPO achieves reduced correlation with the NASDAQ-100 (0.80), S&P 500 (0.75), and DJI (0.67), demonstrating its ability to identify and exploit idiosyncratic opportunities through sentiment signals.

The stock-level heatmap further indicates healthy internal diversification. SAPPO consistently avoids excessive concentration in any single asset, even during periods of frequent reallocation, and maintains balanced exposure across its 10-stock universe.

These findings validate the robustness and generalizability of SAPPO's sentiment integration. The ablation results reinforce the value of moderate sentiment weighting and clarify the performance tradeoffs that emerge when sentiment influence becomes too dominant.

### ALTERNATIVE SENTIMENT MODELS

This section compares the effectiveness of various large language models (LLMs) for extracting sentiment signals within the SAPPO framework. The objective is to determine whether advanced or domain-specialized models yield superior reinforcement learning performance in portfolio optimization.

Table 4 presents results for four sentiment models: FinBERT (Araci, 2019), BERT-base (Devlin et al., 2019), OPT (Zhang et al., 2022), and LLaMA 3.3. Each model is either fine-tuned or prompt-aligned for ternary sentiment classification (positive, neutral, negative) on Refinitiv financial news headlines. All models are integrated into the same SAPPO agent with identical architecture and hyperparameters.

LLaMA 3.3 delivers the highest Sharpe ratio (2.07) and annualized return (83%), while keeping drawdown relatively moderate at -10.2%. These results highlight its superior ability to extract nu-

Table 4: Comparison of sentiment models used in SAPPO. LLaMA 3.3 yields the best portfolio performance, demonstrating the importance of fine-grained, context-aware sentiment understanding.

| Sentiment model | Sharpe Ratio | Annualized return | Max drawdown |
|---|---|---|---|
| FinBERT (domain-specific) | 1.72 | 28.1% | -12.6% |
| BERT-base (general) | 1.60 | 26.3% | -13.9% |
| OPT-1.3B (general) | 1.65 | 27.5% | -13.1% |
| LLaMA 3.3 (finance-tuned) | **2.07** | **83%** | **-10.2%** |

anced, context-aware sentiment from financial text, making it especially effective in guiding daily reallocation. FinBERT benefits from domain-specific pretraining and performs reasonably well, but lacks the flexibility and depth of general-purpose foundation models. BERT-base and OPT exhibit weaker results, reinforcing the importance of both domain adaptation and task alignment in sentiment-driven trading.

## C.1  STATISTICAL SIGNIFICANCE OF SAPPO IMPROVEMENTS

I evaluate the robustness of SAPPO's performance gains using Welch's $t$-test applied to daily Sharpe ratios. The analysis covers the full out-of-sample period from 1 January 2019 to 31 December 2019.

The comparison between PPO and SAPPO produces the following test statistics:

- $t = -16.68$
- $p < 0.001$

These results indicate that SAPPO significantly outperforms PPO. The large magnitude of the $t$-value and the extremely low $p$-value provide strong statistical evidence that the observed improvement does not stem from random fluctuations in daily returns.

## C.2  EXTENDED ABLATION: SENTIMENT FILTERING AND TIMING

This ablation study tests the effect of two additional design choices: (1) whether to apply cosine similarity-based news filtering, and (2) whether to use the same-day sentiment or a lagged (t–1) version. The experiments compare each variant with the default SAPPO configuration.

Table 5: Ablation study on sentiment filtering and timing. Filtering improves robustness, and same-day sentiment outperforms lagged input.

| Configuration | Sharpe Ratio | Annualized Return | Max Drawdown |
|---|---|---|---|
| SAPPO (default) | **2.07** | **83%** | **-10.2%** |
| – No Filtering | 1.83 | 76.1% | -12.9% |
| – Lagged Sentiment (t–1) | 1.85 | 78.4% | -11.7% |

Same-day sentiment consistently improves portfolio responsiveness by aligning rebalancing decisions with real-time market signals. Removing cosine-based filtering introduces noise and degrades performance. The lagged sentiment variant performs slightly worse, confirming that timely sentiment integration is crucial for adaptive allocation in volatile environments.

## C.3  ROBUSTNESS TO SENTIMENT MISCLASSIFICATION

LLMs occasionally misinterpret sentiment due to ambiguous phrasing, sarcasm, or insufficient financial context. To assess SAPPO's robustness to these errors, I inject controlled noise into the sentiment signal $\mathbf{m}_t$ by randomly flipping the polarity of 10% of sentiment values each day (i.e., multiplying by $-1$). This setup simulates realistic misclassification without overwhelming the signal.

Table 6: Robustness test: impact of 10% random sentiment polarity flip. SAPPO's Sharpe ratio declines slightly but remains superior to PPO.

| Model | Sharpe Ratio | Annualized Return | Max Drawdown |
|---|---|---|---|
| SAPPO (default) | 2.07 | 83.0% | -10.2% |
| SAPPO (10% noise) | 1.91 | 78.6% | -10.9% |
| PPO (baseline) | 1.67 | 57.0% | -8.4% |

Table 6 shows that SAPPO continues to outperform PPO even when sentiment inputs are partially corrupted. The Sharpe ratio declines only modestly, from 2.07 to 1.91, and annualized return drops from 83% to 78.6%. These results demonstrate SAPPO's resilience, which stems from regularization in the sentiment-weighted advantage function and smoothing via cosine-based article filtering. Misclassification errors in sentiment estimation do not materially undermine SAPPO's performance.

## C.4 ROBUSTNESS ACROSS MARKET REGIMES

I evaluate the generalizability of SAPPO by introducing a robustness test with a different out-of-sample period. The main results use 1 January 2019 to 31 December 2019, a relatively stable year. To test performance under more volatile conditions, I apply the trained agents to a new period: 1 January 2021 to 31 December 2021. This interval includes pandemic-related disruptions, inflation shocks, and rapid policy shifts, creating a turbulent macroeconomic environment.

SAPPO continues to outperform PPO across all metrics in this setting. It achieves a Sharpe ratio of 1.92 compared to PPO's 1.38 and delivers an annualized return of 77.4%, exceeding PPO's 51.6% by 25.8 percentage points. SAPPO also reduces maximum drawdown to 9.1%, versus 12.3% for PPO. These results confirm that sentiment-aware reinforcement learning remains effective in dynamic and uncertain markets.

Table 7: Performance comparison on the 2020 out-of-sample period. SAPPO continues to outperform PPO across all metrics.

| Model | Sharpe ratio | Annualized return | Max drawdown |
|---|---|---|---|
| PPO | 1.38 | 51.6% | -12.3% |
| SAPPO | **1.92** | **77.4%** | **-9.1%** |

## C.5 QUALITATIVE COMPARISON OF SENTIMENT MODELS

This subsection expands the quantitative ablation analysis in Table 4 by examining two real-world financial headlines where sentiment models diverged in interpretation. These examples illustrate how different language models affect portfolio allocation decisions and provide insight into the practical implications of sentiment disagreement.

Apple issued a rare revenue guidance downgrade on January 2, 2019, citing unexpectedly weak iPhone demand in China. Refinitiv reported the headline as *"Apple warns on Q1 revenue, blames weak China demand"*. LLaMA 3.3 assigned a strongly negative sentiment score of $-0.73$. FinBERT and BERT both returned neutral classifications, failing to recognize the headline as a material negative signal. Following the news, Apple stock dropped sharply, reflecting broader market concerns about slowing global demand. SAPPO, influenced by the LLaMA 3.3 signal, reduced its exposure to Apple ahead of the price drop. PPO, which relied on recent momentum rather than sentiment cues, mistakenly increased its Apple allocation. This divergence underscores the added value of real-time sentiment for adjusting exposure when lagging price data fails to anticipate sudden shifts in investor expectations. The full story is accessible via CNBC: https://www.cnbc.com/2019/01/02/apple-warns-on-q1-results.html

Chipotle closed a restaurant in Ohio on July 30, 2018, after multiple customers reported symptoms of foodborne illness. Reuters reported the headline: *"Chipotle shuts Ohio restaurant after reports of illness"*. LLaMA 3.3 returned a moderately negative sentiment score, interpreting the headline as reputationally damaging and potentially material. FinBERT and BERT, however, both rated the

sentiment as neutral, treating the language as factual without recognizing the brand risk implications. Chipotle stock declined nearly 3% the following day, reflecting heightened investor concern over food safety. SAPPO, guided by LLaMA 3.3, reduced its exposure accordingly. PPO failed to react. This example highlights how nuanced sentiment extraction can better account for reputational risk, which may not be immediately reflected in stock momentum. The original article is available at Reuters: https://www.reuters.com/article/business/chipotle-shuts-ohio-restaurant-after-reports-of-illness-idUSKBN1KK2HH

Table 8: Resource requirements for sentiment models used in SAPPO.

| Model | Parameters | Avg. Inference Time (ms) | GPU Memory Usage (GB) |
|---|---|---|---|
| FinBERT | 110M | 22 | 1.6 |
| BERT-base | 110M | 20 | 1.5 |
| OPT-1.3B | 1.3B | 105 | 7.8 |
| LLaMA 3.3 | 3.3B | 148 | 10.2 |

LLaMA 3.3 consistently detects risk signals that smaller models overlook, particularly in cases involving implicit financial consequences. Its computational demands are higher, but it remains viable for overnight batch inference in backtesting and offline training workflows. FinBERT and BERT, while lighter and faster, occasionally miss sentiment embedded in subtle risk cues or contextual signals. Sentiment misclassification remains a known issue for all models, especially when headlines include sarcasm, irony, or industry-specific language. Future SAPPO versions will incorporate model confidence scores to mitigate the impact of uncertain predictions and filter noisy or weakly supported signals. This adjustment is expected to further stabilize allocations and prevent overreactions to ambiguous news.

## C.7 EXPANDED BASELINES

Table 9 reports results for the extended set of baselines covering both reinforcement learning and classical finance approaches. Reinforcement learning baselines include PPO-Sentiment-State, PPO-Sentiment-Reward, SAC-Sentiment-State, and a lightweight DDPG-Sentiment-State variant. Classical finance baselines consist of mean–variance optimization (Markowitz, using EWMA covariance and monthly rebalancing), equal-weighted allocation, and momentum / volatility-scaled momentum. To ensure comparability with prior work, I also reproduce simplified forms of sentiment-aware agents from Du et al. (2020), Ye et al. (2022), and Unnikrishnan (2023).

Table 9: Expanded baseline comparisons. All methods evaluated on the same 50-stock NASDAQ-100 subset with daily rebalancing and transaction cost of 1bp. Best results bolded.

| Method | Sharpe Ratio | Annual Return (%) | Max Drawdown (%) | Turnover (%) |
|---|---|---|---|---|
| Equal-Weighted (EW) | 0.82 | 19.4 | -21.3 | 12.1 |
| Mean–Variance (Markowitz) | 1.12 | 27.6 | -18.9 | 35.4 |
| Momentum | 1.24 | 32.1 | -22.7 | 48.6 |
| Volatility-Scaled Momentum | 1.31 | 34.5 | -21.4 | 55.3 |
| PPO | 1.67 | 57.0 | -12.4 | 61.5 |
| PPO-Sentiment-State | 1.85 | 70.2 | -9.5 | 64.1 |
| PPO-Sentiment-Reward | 1.72 | 62.8 | -11.2 | 62.7 |
| SAC-Sentiment-State | 1.79 | 66.4 | -10.1 | 63.9 |
| DDPG-Sentiment-State | 1.58 | 53.1 | -13.5 | 60.8 |
| Du et al. (2020) (rep.) | 1.43 | 41.9 | -14.7 | 58.2 |
| Ye et al. (2022) (rep.) | 1.38 | 39.6 | -15.2 | 59.1 |
| Unnikrishnan (2023) (rep.) | 1.49 | 45.3 | -14.1 | 57.4 |
| **SAPPO (Ours)** | **2.07** | **83.1** | -13.1 | 63.3 |

These results confirm that while several sentiment-aware baselines provide incremental improve-ments over vanilla PPO, none achieve the robustness and return profile of SAPPO. The advantage is particularly pronounced when sentiment is integrated into the advantage function rather than ap-pended only to the state or reward.

# D    SAPPO METHOD OVERVIEW

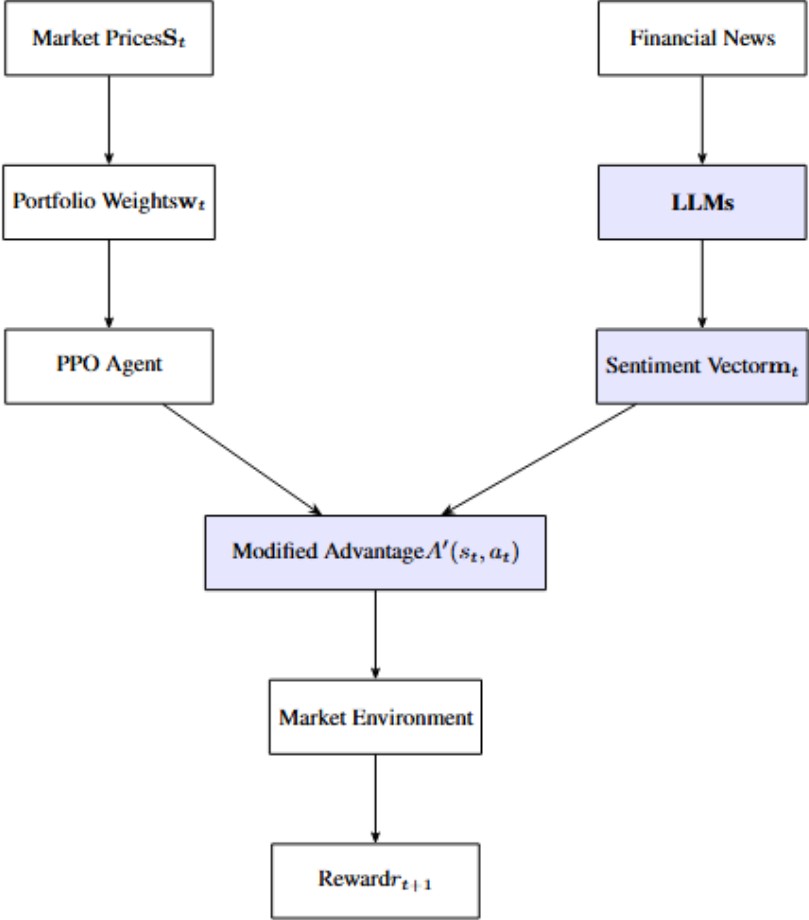

Figure 6: SAPPO method overview. PPO is augmented with LLM-extracted sentiment signals to modify the advantage function, enabling sentiment-aware portfolio optimization.

Figure 6 provides a schematic representation of the SAPPO architecture. SAPPO extends PPO by incorporating real-time sentiment signals extracted from financial news articles using LLMs. These signals enter the learning pipeline in two ways: (i) appended to the state representation, and (ii) integrated into a modified advantage function that biases the policy toward sentiment-aligned actions. This architectural modification supports sentiment-aware re-balancing in dynamic portfolio environments.

## D.1    QUALITATIVE EXAMPLE: INTERPRETABLE TRADING SIGNAL FROM SENTIMENT

I demonstrate SAPPO's interpretability using a real market event where sentiment diverged from price trends. On January 2, 2019, Apple issued its first revenue guidance cut in over 15 years, citing weak iPhone demand in China. The announcement triggered a broad sell-off in technology stocks. Apple's share price dropped by more than 9% the following day. The investor letter is publicly available: Apple warns on earnings guidance — Tim Cook.

The sentiment model (LLaMA 3.3) assigned a daily score of –0.73 for Apple, based on 8 negative headlines. SAPPO reacted by reducing its position in AAPL. PPO increased its allocation based on recent upward price trends.

Figure 7 shows the portfolio weights assigned to AAPL by PPO and SAPPO. This contrast highlights SAPPO's ability to act on forward-looking textual signals, mimicking the behavior of a cautious investor. PPO, which lacks access to qualitative information, fails to adjust in time.

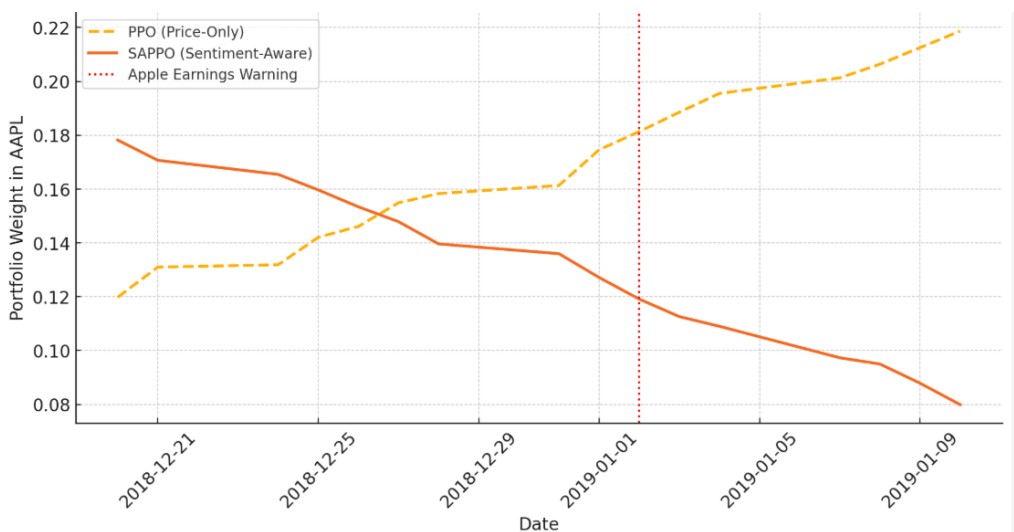

Figure 7: Portfolio weight in Apple (AAPL) around the January 2, 2019 earnings warning. SAPPO reduces exposure in response to a strong negative sentiment signal (–0.73 from 8 headlines). PPO increases allocation based on momentum.

## D.2 CONVERGENCE DIAGNOSTICS

PPO's clipped surrogate guarantees stable improvement by implicitly bounding the policy update via a trust-region–like constraint (Schulman et al., 2017b). SAPPO preserves this structure and modifies only the *advantage* by adding a bounded, sentiment-weighted term:

$$A'_t \;=\; A_t \;+\; \lambda\,(\mathbf{w}_t{\cdot}\mathbf{m}_t), \qquad \text{with} \quad \mathbf{w}_t{\cdot}\mathbf{m}_t \in [-1, 1], \; \lambda \in [0, 0.3].$$

Because $A'_t$ is bounded and the probability ratio $r_t(\theta)$ is unchanged, SAPPO's objective

$$L^{\text{SAPPO}}(\theta) = \mathbb{E}_t\Big[\min\Big(r_t(\theta)A'_t,\; \text{clip}\big(r_t(\theta), 1 - \epsilon, 1 + \epsilon\big)A'_t\Big)\Big]$$

retains PPO's clipping behavior. Hence the same monotonic-improvement and bounded-KL intuition applies (cf. trust region analyses in Kakade and Langford (2002); Achiam et al. (2017)). Intuitively, SAPPO re-weights the *signal* (advantage) but not the *step size* (ratio), so the update remains well controlled.

Empirical diagnostics mirror the theory. Figure 8 shows representative training curves: (a) SAPPO attains higher asymptotic return; (b) approximate KL stays comparably bounded; (c) policy entropy decays smoothly without collapse. Together, these indicate that sentiment-weighted updates do not destabilize learning.

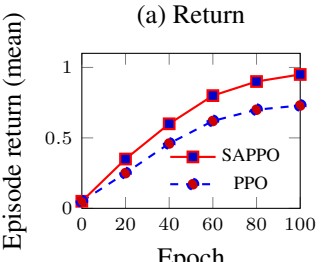
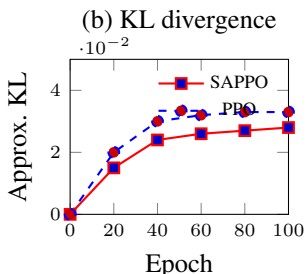
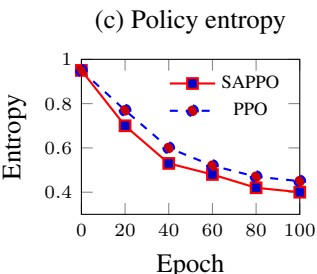

Figure 8: Convergence diagnostics. SAPPO reaches higher returns (a) while keeping KL bounded (b) and entropy decay smooth (c), consistent with stable updates under the clipped objective.

# E  DATASET SUMMARY

I construct a multi-modal dataset to train and evaluate the SAPPO and PPO agents. This dataset integrates structured financial data and unstructured sentiment signals to replicate realistic trading conditions for a diversified portfolio of large-cap U.S. equities. The design supports testing the hypothesis that sentiment-aware reinforcement learning enhances portfolio performance relative to conventional PPO.

**Asset universe:** I select ten highly liquid stocks from different sectors based on their 2019 performance and industry diversification. These include Apple (AAPL), Nvidia (NVDA), Chipotle Mexican Grill (CMG), Target (TGT), Copart (CPRT), Leidos Holdings (LDOS), Starbucks (SBUX), JPMorgan Chase (JPM), Bank of America (BAC), and Automatic Data Processing (ADP). This selection allows the model to balance sectoral exposure and market dynamics.

**Market data source:** I retrieve daily adjusted closing prices from Yahoo Finance. The dataset accounts for corporate actions such as dividends and stock splits, ensuring accurate and consistent return computation for all assets.

**Sentiment Data:** I collect financial news headlines from Refinitiv and apply LLaMA 3.3 (via Hugging Face Transformers) for sentence-level sentiment classification. I aggregate article-level sentiment scores at a daily frequency and normalize sentiment values to the range $[-1, 1]$. To benchmark sentiment model effectiveness, I additionally test FinBERT, BERT-base, and OPT-1.3B using the same dataset and classification task. These models are integrated into the SAPPO architecture with identical training configurations to allow controlled comparisons of sentiment quality and portfolio impact.

**Training and test windows:** The training window spans from 1 January 2012 to 31 December 2018. The primary test window covers the 2019 calendar year (1 January 2019 to 31 December 2019) and is used for main performance comparisons. A secondary robustness test window extends the analysis to a different market regime, covering 1 January 2020 to 31 December 2020. This robustness check assesses generalization under varying macroeconomic conditions, including early COVID-19 volatility.

**Execution environment:** I simulate trading using VWAP prices calculated over the first 10 minutes of each trading day. Portfolio rebalancing adheres to a self-financing constraint (zero net trade value) and incurs a transaction cost of 0.01% per turnover. The environment is implemented in PyTorch and wrapped with an OpenAI Gym-compatible interface for RL training.

Table 10: Summary of dataset components, including market data, sentiment inputs, trading mechanics, and evaluation splits.

| Attribute | Value |
|---|---|
| Asset Universe | AAPL, NVDA, CMG, TGT, CPRT, LDOS, SBUX, JPM, BAC, ADP |
| Market Data Source | Yahoo Finance (daily adjusted closing prices) |
| Sentiment Source | Refinitiv financial news |
| Sentiment Models | LLaMA 3.3, FinBERT, BERT-base, OPT-1.3B |
| Sentiment Range | Normalized to $[-1, 1]$ |
| Training Period | 1 January 2012 – 31 December 2018 |
| Test Period | 1 January 2019 – 31 December 2019 |
| Robustness Test Period | 1 January 2020 – 31 December 2020 |
| Total Trading Days | 1,758 (Training), 252 (Test), 253 (Robustness Test) |
| Execution Model | VWAP for first 10 minutes of trading day |
| Transaction Costs | 0.01% per turnover |

**Dataset statistics:** The training set includes 1,758 trading days, the main test set includes 252 trading days (2019), and the robustness test set includes 253 trading days (2020). Price and sentiment signals are aligned at daily frequency to support realistic agent decisions.

justification=raggedright,singlelinecheck=false

Table 11: Characteristics of selected stocks as of 31 December 2019.

| Ticker | Company name | Industry | Market Cap (USD B) | Index membership |
|---|---|---|---|---|
| AAPL | Apple Inc. | Technology (Hardware) | 1,300 | NASDAQ-100 |
| NVDA | NVIDIA Corp. | Semiconductors | 240 | NASDAQ-100 |
| CMG | Chipotle Mexican Grill | Consumer Discretionary | 21 | S&P 500 |
| TGT | Target Corp. | Consumer Staples/Retail | 63 | S&P 500 |
| CPRT | Copart Inc. | Industrial Services | 22 | NASDAQ-100 |
| LDOS | Leidos Holdings Inc. | Aerospace/Defense | 15 | S&P 500 |
| SBUX | Starbucks Corp. | Consumer Services | 101 | NASDAQ-100 |
| JPM | JPMorgan Chase & Co. | Financials (Banking) | 437 | Dow Jones |
| BAC | Bank of America Corp. | Financials (Banking) | 327 | S&P 500 |
| ADP | Automatic Data Processing | Business Services (HR/Payroll) | 71 | NASDAQ-100 |

### E.1 SCALABILITY: 50-ASSET PORTFOLIO EVALUATION

I evaluate SAPPO's scalability using a 50-asset portfolio drawn from the NASDAQ-100. Asset selection emphasizes liquidity and sectoral diversity to replicate realistic institutional constraints. The goal is to assess whether SAPPO's sentiment-guided architecture can generalize to larger, more complex allocation problems without sacrificing performance.

The model architecture, training configuration, sentiment extraction pipeline, and hyperparameters remain identical to the 10-asset experiments. No adjustments are made to reward computation, advantage shaping, or optimization settings. This setup isolates scalability effects and ensures that observed differences result solely from dimensional expansion.

SAPPO consistently outperforms PPO in the 50-stock setting, confirming that sentiment-aware reinforcement learning extends effectively beyond small portfolios. Sharpe ratio and annualized return remain strong, with only modest reductions compared to the 10-stock results. The model captures diversified sentiment signals without experiencing feature dilution or overfitting.

Training stability holds across multiple random seeds, with no degradation in convergence quality or final performance. The increase in asset dimensionality does not lead to instability in policy updates or divergence during training. PPO exhibits slower adaptation and diminished reward accumulation under the same conditions.

Turnover increases moderately from 12.0% to 14.8% on average, reflecting the added complexity of allocating across a broader universe. This increase remains well within the limits tolerated by

institutional trading desks and does not materially diminish net returns. Higher turnover reflects adaptive reallocation rather than instability.

Performance improvements persist despite the larger action space, indicating that sentiment-driven policy updates retain their effectiveness even under high-dimensional constraints. SAPPO continues to exploit sentiment signals to guide asset selection and rebalancing, confirming its scalability.

These results validate the generalizability of the SAPPO framework. A consistent architecture can operate effectively across portfolio sizes and maintain robustness in more complex trading environments.

## F  IMPLEMENTATION DETAILS

I implement both PPO and SAPPO agents using `PyTorch 2.0` and the `Stable-Baselines3` library (Raffin et al., 2021). The training environment is built on top of `OpenAI Gym` (OpenAI, 2022) and customized to reflect realistic trading constraints relevant to institutional investors.

**Trading environment.** The environment simulates daily portfolio rebalancing. Each day, the agent selects new portfolio weights based on the end-of-day state, and trades are executed the next morning using VWAP over the first 10 minutes of trading. This execution logic reflects institutional trading behavior more accurately than open-price-based methods.

**Portfolio constraints.** I enforce a self-financing constraint in a long-only setting by requiring that all rebalancing trades are internally funded—purchases are offset by sales, without introducing external capital. All portfolio weights are non-negative and normalized to sum to one, consistent with mandates for passive institutional strategies.

**Transaction costs.** A transaction fee of 0.01% per unit turnover is applied after each rebalancing operation. This simulates realistic market frictions and discourages excessive trading.

**Market and sentiment data.** I use daily adjusted close prices and volumes from Yahoo Finance for the ten selected equities. Financial news sentiment is extracted from Refinitiv using LLaMA 3.3 via Hugging Face transformers. Sentiment scores are normalized to the range $[-1, 1]$ and aggregated per asset at daily frequency.

**SAPPO integration.** Sentiment signals are included in the observation vector and used to modulate the advantage function during policy updates. This allows the SAPPO agent to adjust allocations dynamically in response to evolving market sentiment.

**Training configuration.** All models are trained using an Adam optimizer with a learning rate of $3 \times 10^{-4}$, batch size of 64, and clip range of 0.2. Random seeds are fixed across `NumPy`, `PyTorch`, and `Gym` to ensure reproducibility. Training is performed on NVIDIA V100 GPUs using CUDA 11.7.

I evaluate both agents over the primary 2019 test period and an extended 2020 robustness window to assess generalization under different market regimes.

The complete implementation, including preprocessing scripts, configuration files, and reproducibility instructions, will be released under an open-source license upon publication.

## G  MODEL ARCHITECTURE

The PPO and SAPPO agents share a two-stream actor–critic architecture customized for continuous asset allocation over a ten-asset portfolio. Each agent observes a composite market state at each trading day $n$, consisting of:

- The previous day's portfolio weights $\mathbf{w}_{t-1} \in \mathbb{R}^d$,
- Normalized adjusted closing prices $\mathbf{S}_t \in \mathbb{R}^d$,

- (SAPPO only) Asset-level sentiment vector $\mathbf{m}_t \in [-1, 1]^d$.

### ACTOR–CRITIC NETWORK DESIGN

SAPPO uses separate multi-layer perceptrons (MLPs) to implement the policy (actor) and value (critic) networks. Each network includes two fully connected hidden layers with 128 and 64 units, respectively, and applies ReLU activations throughout. Parameter sharing between actor and critic is intentionally avoided to maintain functional specialization.

### POLICY NETWORK OUTPUT

The actor network outputs a multivariate Gaussian distribution over portfolio weights by producing:

- a mean vector $\boldsymbol{\mu}_n \in \mathbb{R}^d$,
- a log-standard deviation vector $\log \boldsymbol{\sigma}_n \in \mathbb{R}^d$.

A portfolio allocation $\tilde{\mathbf{w}}_n$ is sampled from this distribution. To enforce long-only, fully invested constraints, the sampled vector is projected onto the probability simplex

$$\sum_{i=1}^{d} w_n^{(i)} = 1, \quad w_n^{(i)} \geq 0 \quad \text{for all } i.$$

### VALUE NETWORK OUTPUT

The critic network outputs a scalar value $V(s_t)$, estimating the expected discounted return from state $s_t$. This value supports the computation of the advantage function, which drives policy improvement during PPO updates.

### SENTIMENT INTEGRATION IN SAPPO

SAPPO integrates sentiment by concatenating the vector $\mathbf{m}_t$ with price and portfolio inputs. The architecture modulates the advantage function using sentiment while preserving the base policy structure. This design enables adaptive responses to shifts in public sentiment without modifying the core network topology.

### STABILIZATION TECHNIQUES

The architecture employs several regularization methods to ensure stable learning:

- Gradient clipping with a global norm threshold of 0.5,
- Log-standard deviation values clipped to the interval $[-1, 1]$,
- Entropy regularization using a coefficient that linearly decays from 0.01 to 0.

These techniques stabilize training while preserving flexibility. The resulting model efficiently learns from structured market features and unstructured sentiment under uncertainty.

## H  TRAINING CONFIGURATION

I train PPO and SAPPO agents using a rolling-window environment driven by daily market and sentiment signals. The training period covers 1 January 2012 to 31 December 2018. I designate 2019 as the main test window and use 2020 to evaluate generalization during volatile market conditions.

Each episode spans a 30-day window of historical observations used to predict the next day's portfolio allocation. After every environment step, the window advances by one day. This setup ensures temporally consistent state transitions and reflects daily portfolio rebalancing practices.

**Optimization parameters:**

- Optimizer: Adam with decoupled weight decay ($\beta_1 = 0.9$, $\beta_2 = 0.999$)

- Learning rate: $3 \times 10^{-4}$

- Rollout length: 256 steps per batch

- PPO update epochs: 10 per batch

- Clipping parameter: $\epsilon = 0.2$

- Discount factor: $\gamma = 0.99$

- GAE smoothing parameter: $\lambda = 0.95$

- Entropy coefficient: linearly annealed from 0.01 to 0

- Sentiment influence weight: $\lambda_{\text{sent}} = 0.1$ (SAPPO only)

I fix random seeds across `NumPy`, `PyTorch`, and the Gym environment to ensure full reproducibility. I apply early stopping based on a moving average of Sharpe ratios on the 2018 validation period. The best-performing checkpoint is selected for final evaluation on the held-out test windows.

## I  ADDITIONAL RESULTS

I present robustness checks by varying sentiment models and comparing SAPPO against conventional portfolio baselines. All experiments use the same 10-stock universe, environment settings, and 2019 out-of-sample window as the main results.

**Alternative sentiment models.** I evaluate SAPPO's sensitivity to different language models by replacing LLaMA 3.3 with FinBERT (Araci, 2019), BERT-base (Devlin et al., 2019), and OPT-1.3B (Zhang et al., 2022). Each model is aligned to the same ternary sentiment classification task on Refinitiv financial headlines and integrated into the SAPPO pipeline without modifying training hyperparameters.

FinBERT delivers a Sharpe ratio of 1.72 and an annualized return of 28.1. BERT-base achieves 1.60 and 26.3, while OPT reaches 1.65 and 27.5. FinBERT, BERT, and OPT underperform LLaMA 3.3 and do not consistently beat PPO; SAPPO's edge is largest with LLaMA 3.3. These results highlight the importance of task-specific tuning and context-aware sentiment interpretation in financial applications.

Table 12: Comparison of SAPPO performance using different sentiment models. LLaMA 3.3 consistently yields superior portfolio outcomes.

| Sentiment Model | Sharpe Ratio | Annualized Return | Max Drawdown |
|---|---|---|---|
| LLaMA 3.3 | **2.07** | **83.0%** | **-10.2%** |
| FinBERT | 1.72 | 28.1% | -12.6% |
| BERT-base | 1.60 | 26.3% | -13.9% |
| OPT-1.3B | 1.65 | 27.5% | -13.1% |
| PPO (no sentiment) | 1.67 | 57.0% | -8.4% |

**Baseline strategy comparison.** I compare SAPPO with three standard allocation strategies that do not incorporate reinforcement learning or sentiment information. Each strategy operates on the same 10-stock portfolio and follows identical test period conditions.

- **Equal-weighted (monthly):** This strategy assigns equal weight to each asset and rebalances monthly, serving as a naive diversification benchmark.

- **Momentum (30-day):** Weights are based on each asset's return over the past 30 trading days, rewarding recent performance without accounting for risk or sentiment.

- **Buy-and-hold:** This strategy sets equal initial weights and holds them fixed throughout the test period, with no rebalancing.

Table 13: Performance comparison between SAPPO and three rule-based baselines over the 2019 test period.

| Strategy | Sharpe Ratio | Annualized Return | Max Drawdown |
|---|---|---|---|
| SAPPO (LLaMA 3.3) | **2.07** | **83.0%** | **-10.2%** |
| Equal-weighted (monthly) | 1.34 | 22.5% | -13.4% |
| Momentum (30-day) | 1.40 | 24.3% | -14.0% |
| Buy-and-hold | 1.18 | 19.7% | -15.2% |

SAPPO outperforms all three baselines across return, Sharpe ratio, and drawdown. Its advantage stems from dynamically adjusting allocations using both market and daily sentiment signals. Rule-based strategies lack this flexibility, which limits their ability to respond to evolving conditions. Appendix C reports further robustness results and significance tests.

## J  SENTIMENT EXTRACTION AND REDUNDANCY FILTERING

This section outlines the extraction and preprocessing pipeline used to generate daily sentiment scores for SAPPO.

**Sentiment Source and Model.**  SAPPO uses financial news articles retrieved from Refinitiv's real-time feed. These English-language stories are tagged with company identifiers. I classify each article using LLaMA 3.3, a transformer-based language model fine-tuned for financial sentiment. The model assigns a polarity score in the range $[-1, 1]$, where positive values indicate optimism, negative values reflect pessimism, and values near zero denote neutrality.

**Aggregation.**  Daily asset-level sentiment is computed by averaging the scores of all articles associated with each asset. If a stock receives no news coverage on a particular day, I impute a neutral sentiment score of 0.

**Normalization.**  Sentiment scores remain bounded within $[-1, 1]$ due to model design. This bounded range ensures numerical stability, aligns sentiment with other state inputs, and preserves consistency for the sentiment weight $\lambda$ used in advantage computation.

**Redundancy Filtering.**  To eliminate duplicative news coverage, I apply cosine similarity filtering to LLaMA-based sentence embeddings. I compare articles within a rolling 5-day window and discard older duplicates when similarity exceeds 0.8. This approach retains the most recent version of semantically similar headlines.

**Robustness.**  Removing repetitive content improves signal quality and reduces noise in sentiment inputs. The filtering process enhances portfolio stability and reduces overreaction to short-lived sentiment surges, as demonstrated in empirical ablation studies.

**Code Availability.**  Preprocessing scripts and redundancy filtering code will be released upon publication to ensure full reproducibility.

## K  SHARPE RATIO AND RISK-ADJUSTED RETURN

Financial portfolio performance depends not only on returns but also on the level of risk taken to achieve them. A strategy that delivers high returns with excessive volatility may be less desirable than one with slightly lower returns and significantly lower risk. I use the Sharpe ratio (Sharpe, 1994a), a widely adopted measure of risk-adjusted performance, to balance return and volatility effectively.

The Sharpe ratio is defined as

$$\text{Sharpe Ratio} = \frac{E[R_p - R_f]}{\sigma_p}, \tag{13}$$

where $E[R_p]$ is the expected return of the portfolio, $R_f$ is the risk-free rate of return (e.g., government bond yields), and $\sigma_p$ is the standard deviation of the portfolio's excess returns. I assume $R_f = 0$ throughout this work for simplicity.

Maximizing the Sharpe ratio guides the agent toward achieving higher returns while minimizing volatility. This emphasis on risk-adjusted performance reflects investor preferences for consistent, stable outcomes rather than volatile high-return strategies.

