# OpenReview forum: "Sentiment-weighted advantage updates for portfolio optimization with reinforcement learning"
_ICLR.cc/2026/Conference — ICLR 2026 Conference Withdrawn Submission_

### Official Review · Reviewer_iLLX · 2025-10-17

**Soundness:** 2
**Presentation:** 1
**Contribution:** 1
**Rating:** 2
**Confidence:** 4

**Summary:**

The author introduces a new method of reinforcement learning based algorithm to train an agent for making decisions on portfolio investment. Different with prior work, in this work, sentiment analysis results from news into both the state representation and the objective reward in the PPO algorithm. Significant gain from using the proposed methodology in the expected return is observed against baselines.

**Strengths:**

- Incorporation of sentiment analysis results in the advantage estimation is an interesting exploration.
- The experiments are comprehensive, and the choice of each component as well as the alternative combinations of modules are well justified through ablation studies. The performance gain achieved by the proposed SAPPO is consistent across all reported experiments.

**Weaknesses:**

- The methodology overall is a bit incremental, where incorporating sentiment analysis to RL based investment planning was widely explored in prior literature, also as discussed by the author. It would be better to also present the performance of those prior approaches mentioned in paper into the baselines, which will give readers better sense of how this work is placed in the current literature.
- The insights from the study is a bit insufficient, where the expected return/performance is the main focus, but less insights/analysis/observation are conducted on more in-depth aspects, such as to what extent the sentiment information influences the optimal policy learned by the agent? What is the distribution of the sentiment levels? Is the agent's learned policy influence more by the positive or negative news? If the emphasis is on the inclusion of sentiment, is the performance gain consistent across varied RL algorithms (e.g. Q learning, Gradient Policy, DPO, etc.)?
- The sentiment analysis results are incorporated into the advantage estimation, what if the raw LLM embedding is leveraged? Would that way provide the agent more informative signals?
- The presentation quality could be improved. Most figures are not vectorized and the font resolution appears low.

**Questions:**

My questions are stated in the Weaknesses above. Having these additional aspects being analyzed or discussed will improve the depth of the proposed method.

---

> ### Author Response · Authors · 2025-11-12
>
> I am not able to take this review seriously with conflicting comments in the weaknesses and overall comments. If the study is comprehensive as the Chatgpt of the reviewer wrote why did he mark 2? Vague comments with no merritts. He/She speaks about insights but has he looked at 25 pages of Appendix?

---

### Official Review · Reviewer_5DoB · 2025-10-27

**Soundness:** 2
**Presentation:** 2
**Contribution:** 2
**Rating:** 2
**Confidence:** 4

**Summary:**

This paper introduces sentiment-augmented PPO, a RL framework for portfolio optimization that integrates financial news sentiment into both the state representation and the policy update mechanism. Experiments on multiple US stock datasets demonstrate its SOTA performance.

**Strengths:**

1. The core sentiment-weighted advantage function seems a simple yet efficient trick for portfolio optimization.
2. The proposed method shows robust performance with good generalization ability.

**Weaknesses:**

1. This paper is more like a technical report rather than a research paper for top conferences. The proposed method is a bit trivial with limited technical contribution for me.
2. Since sentiment is the key point for this work, more experiments with sentiment derived from other text data (e.g., social media, analyst reports) can make this work stronger.
3. The backtest environment is not realistic without the consideration of slippage and so on.
4. How practical is the proposed methods for intra-day setting or live trading? Current evaluation only focuses on day-level settings.

**Questions:**

Please see weakness above.

---

> ### Author Response · Authors · 2025-11-12
>
> I am refusing to respond to the reviewer. The reviewer has not read the paper.

---

### Official Review · Reviewer_VJvd · 2025-10-29

**Soundness:** 2
**Presentation:** 2
**Contribution:** 2
**Rating:** 4
**Confidence:** 4

**Summary:**

This paper introduces Sentiment-Augmented PPO, a reinforcement learning framework that integrates investor sentiment directly into the policy update process of PPO. Unlike previous approaches that inject sentiment as an auxiliary feature or use it for reward shaping, SAPPO modifies the advantage function itself through a sentiment-weighted mechanism, where asset-level sentiment scores modulate policy gradients during learning. This integration aims to create behaviorally informed trading agents that adjust strategies based on collective market sentiment. Using Refinitiv news data processed by a fine-tuned LLaMA 3.3 model, sentiment vectors are appended to the state and used in the advantage computation.

**Strengths:**

1. By incorporating sentiment into the advantage computation (Eq. 2), the paper moves beyond prior sentiment-as-feature paradigms and introduces a novel mechanism for modulating policy gradients in a behaviorally grounded way. The approach remains mathematically simple yet produces substantial empirical gains in both Sharpe ratio and cumulative return.

2. The empirical validation is thorough, including diverse baselines (e.g., PPO-Sentiment-State, PPO-Sentiment-Reward, SAC-Sentiment-State), robustness tests on larger portfolios, and expanded time periods

**Weaknesses:**

1. The computational cost associated with sentiment extraction (LLaMA 3.3 inference, cosine filtering, daily aggregation) is only described qualitatively as efficient, but no quantitative training-time or resource comparison is provided. For practical deployment in high-frequency trading, this detail is crucial.

2. SAPPO exhibits increased turnover (12% vs. 3.5% for PPO) and trading cost (3% of annual portfolio value), which could reduce net profit under realistic transaction fees or liquidity constraints.

3. The paper’s interpretability analysis remains surface-level: while sentiment influences are discussed qualitatively, there is no deeper causal attribution or visualization connecting sentiment shifts to specific trading actions.

**Questions:**

1. How does sentiment weighting affect PPO’s theoretical convergence?
Can the authors provide a formal derivation or stability proof showing that the sentiment-weighted advantage update (Eq. 2) preserves PPO’s monotonic improvement property?

2. Can turnover and transaction costs be further mitigated?
Given SAPPO’s higher daily turnover, have the authors explored regularization methods (e.g., turnover penalties or entropy constraints) to maintain adaptability while reducing trading frequency?

3. How robust is the model to sentiment errors or lags?
Since sentiment extraction relies on LLaMA-based NLP models, can SAPPO maintain its advantage if sentiment data is noisy, delayed, or misclassified?

---

### Official Review · Reviewer_qaGM · 2025-11-01

**Soundness:** 2
**Presentation:** 3
**Contribution:** 2
**Rating:** 2
**Confidence:** 4

**Summary:**

This paper proposes Sentiment-Augmented PPO (SAPPO), a simple extension of the Proximal Policy Optimization algorithm for portfolio management that integrates market sentiment into the policy-update step. Instead of adding sentiment as an input feature, SAPPO weights the advantage function by a sentiment score, so that policy updates are scaled up or down depending on whether market sentiment is positive or negative. This modification allows the agent to adapt its risk-taking behavior dynamically without altering PPO’s theoretical structure or stability. Experiments on multiple financial datasets show that SAPPO achieves higher returns and Sharpe ratios than standard PPO and other sentiment-aware baselines, while maintaining robustness across market regimes. The paper argues that this approach is a simple, interpretable, and effective way to fuse qualitative sentiment information with quantitative reinforcement-learning strategies.

**Strengths:**

The paper tackles a clear and timely problem—how to make reinforcement learning for portfolio management more responsive to market sentiment. Its main idea, weighting the PPO advantage by sentiment, is simple, intuitive, and easy to implement. The method keeps PPO’s efficiency and stability while adding useful behavioral awareness. Experiments are thorough and show consistent gains across datasets and conditions. Overall, it’s a practical, well-executed approach with strong empirical results and clear real-world potential.

**Weaknesses:**

While the proposed sentiment-weighted advantage method is simple and effective, it is not fully convincing that this is the best or most principled way to integrate sentiment information into reinforcement learning. Sentiment could be incorporated in several other ways—for example, as input features to the policy or value network, as part of the reward function, or through adaptive risk control mechanisms. Moreover, a classical two-step approach—first predicting future returns using sentiment-enhanced models, then performing portfolio optimization based on those forecasts—remains a natural and well-established alternative. The paper does not clearly explain why direct weighting of the advantage is superior to these more conventional or modular strategies, nor does it provide comparisons against strong sentiment-aware baselines that use such setups. As a result, the contribution, while practical, may appear somewhat incremental or empirically motivated rather than conceptually justified.

**Questions:**

Could the authors elaborate on why weighting the advantage function by sentiment is a more effective or principled integration point than incorporating sentiment through other mechanisms, such as reward shaping, state augmentation, or adaptive risk scaling?

---

> ### Author Response · Authors · 2025-11-12
>
> I am refusing to write a comment for this reviewer as the reviewer is CHATGPT

---

### Note · Authors · 2025-11-20

**Comment:**

Poor review process
Most of the reviewers have used Chatgpt to review the paper

**Withdrawal Confirmation:**

I have read and agree with the venue's withdrawal policy on behalf of myself and my co-authors.